# Perinatal granulopoiesis and risk of pediatric asthma

Benjamin A Turturice[1,2]*, Juliana Theorell[2], Mary Dawn Koenig[3], Lisa Tussing-Humphreys[4], Diane R Gold[5,6], Augusto A Litonjua[7], Emily Oken[8], Sheryl L Rifas-Shiman[8], David L Perkins[9,10]*, Patricia W Finn[1,2,10]*

[1]Department of Microbiology and Immunology, University of Illinois, Chicago, United States; [2]Department of Medicine, Division of Pulmonary, Critical Care, Sleep, and Allergy, University of Illinois, Chicago, United States; [3]Department of Women, Children and Family Health Science, College of Nursing, University of Illinois, Chicago, United States; [4]Department of Medicine and Cancer Center, University of Illinois, Chicago, United States; [5]Channing Division of Network Medicine, Department of Medicine, Brigham and Women's Hospital, Harvard Medical School, Boston, United States; [6]Department of Environmental Health, Harvard T.H. Chan School of Public Health, Boston, United States; [7]Division of Pulmonary Medicine, Department of Pediatrics, University of Rochester, Rochester, United States; [8]Division of Chronic Disease Research Across the Life Course, Department of Population Medicine, Harvard Medical School and Harvard Pilgrim Health Care Institute, Boston, United States; [9]Department of Medicine, Division of Nephrology, University of Illinois, Chicago, United States; [10]Department of Bioengineering, University of Illinois, Chicago, United States

*For correspondence:
bturtu2@uic.edu (BAT);
perkinsd@uic.edu (DLP);
pwfinn@uic.edu (PWF)

Competing interests: The authors declare that no competing interests exist.

**Abstract** There are perinatal characteristics, such as gestational age, reproducibly associated with the risk for pediatric asthma. Identification of biologic processes influenced by these characteristics could facilitate risk stratification or new therapeutic targets. We hypothesized that transcriptional changes associated with multiple epidemiologic risk factors would be mediators of pediatric asthma risk. Using publicly available transcriptomic data from cord blood mononuclear cells, transcription of genes involved in myeloid differentiation was observed to be inversely associated with a pediatric asthma risk stratification based on multiple perinatal risk factors. This gene signature was validated in an independent prospective cohort and was specifically associated with genes localizing to neutrophil-specific granules. Further validation demonstrated that umbilical cord blood serum concentration of PGLYRP-1, a specific granule protein, was inversely associated with mid-childhood current asthma and early-teen $FEV_1/FVCx100$. Thus, neutrophil-specific granule abundance at birth predicts risk for pediatric asthma and pulmonary function in adolescence.

## Introduction

Several risk factors for pediatric asthma can be ascertained in the perinatal period. These risk factors include maternal characteristics (e.g., maternal atopy, maternal body mass index [BMI], race/ethnicity), demographics (e.g., newborn sex), and birth characteristics (e.g., birthweight, gestational age at birth, mode of delivery) (*Bisgaard and Bønnelykke, 2010*). Meta-analyses have provided strong evidence for associations between the variables stated above and risk for pediatric asthma (*Jaakkola et al., 2006*; *Mu et al., 2014*; *Thavagnanam et al., 2008*; *Xu et al., 2014*). Many of these risk factors co-occur (e.g., low birthweight and preterm birth), and it has yet to be discerned whether their imparted risk is mediated through similar biologic processes.

Meta-analyses assessing peripheral blood leukocytes of school-aged children have identified differentially methylated regions proximal, or within, genes specifically transcribed in eosinophils as a common signature in pediatric asthma (*Xu et al., 2018*; *Reese et al., 2019*). These findings are consistent with observations of enhanced T2 inflammatory responses in children with asthma (*Wenzel, 2012*). However, when assessing cord blood mononuclear cell (CBMC) samples, differential methylation did not extend to these and other classically T2-associated loci (*Reese et al., 2019*). Interestingly, some individuals are predisposed at birth to generating T2 responses ex vivo to common asthma triggers (e.g. aeroallergens) or having detectable IgE concentrations in cord blood but neither has been shown to predict asthma later in life (*Schaub et al., 2005*; *Turturice et al., 2017a*; *Shah et al., 2011*). These findings are suggestive of limited prognostication in asthma risk provided by the variation in T2 immunity at birth. Additionally, efforts aimed at modulating immunity in utero and through early life (e.g. vitamin D, probiotics) have failed to demonstrate benefit in the prevention of asthma (*Litonjua et al., 2020*; *Azad et al., 2013*). Further investigation is required to understand the aspects of newborn immunity associated with pediatric asthma.

The neonatal immune system undergoes many developmental changes throughout gestation and early life. Throughout the majority of gestation, fetal hematopoiesis generates mainly lymphoid and erythroid lineages. While the bone marrow capacity to produce all cell lineages increases toward term gestation, the ability to produce myeloid lineages is most pronounced in later gestational ages (*Forestier et al., 1991*; *Glasser et al., 2015*). Further highlighting the unique immunology of the perinatal time period, there are cell populations (e.g. CXCL8-producing T cells, myeloid-derived suppressor cells) that are highly abundant in cord blood that are rapidly depleted over the first week (*Gibbons et al., 2014*; *Olin et al., 2018*; *Rieber et al., 2013*). Importantly, in utero factors, such as preterm birth and gestational hypertension, can impact perinatal hematopoiesis. These events can impair neutrophil abundance and function at birth but return to adult levels within days of birth similar to that of those who did not have such an exposure (*Glasser et al., 2015*; *Olin et al., 2018*; *Schmutz et al., 2008*). This highlights great variability in the early-life hematopoietic composition driven by in utero differences that rapidly converges to a new baseline as the neonates adapts to its new environment.

This variability in immunity the perinatal time period might be reflective of the presence of multiple risk factors for asthma and facilitate a more detailed risk stratification and, ultimately, identification of potential therapeutic targets. Our focus was to determine biologic processes – extending to both transcriptional and serologic levels – associated with pediatric asthma risk that are detectable at birth. We hypothesized that transcriptional changes in CBMCs associated with multiple epidemiologic risk factors would be mediators of pediatric asthma risk. CBMCs have been previously studied with regards to cytokine production, DNA methylation, and outcomes (*Reese et al., 2019*; *Turturice et al., 2017a*; *Lin et al., 1993*; *Ly et al., 2007*; *Turturice et al., 2019*; *den Dekker et al., 2019*), making them ideal candidates for investigation. Here, we identify a novel association between epidemiologic risk, neutrophil-specific granules, and pediatric pulmonary outcomes including childhood asthma.

## Results

### Approach to identify immunologic differences associated with risk for pediatric asthma

We developed an analytic approach to identify genes whose expression in CBMCs are associated with newborns with higher or lower risk for asthma. We conducted a meta-analysis to increase power and generalizability (*Figure 1A,B*). In our approach, we queried NCBI's Gene Expression Omnibus for CBMC microarray datasets that included metadata regarding the demographics and birth characteristics that are risk factors for pediatric asthma. We identified 354 datasets from our original search, of which 17 datasets contained relevant metadata. Of the 17 studies, the most common maternal and neonatal characteristics reported were newborn sex, gestational age at birth, birthweight, and maternal pre-pregnancy BMI (PP BMI) at 69.96%, 51.59%, 30.27%, and 19.32% of samples, respectively. Metadata regarding maternal smoking and mode of delivery (i.e., vaginal vs. cesarean section) was reported for three datasets; however, two of these datasets originated from the same laboratory group and contributed the majority of samples reporting these factors. None of

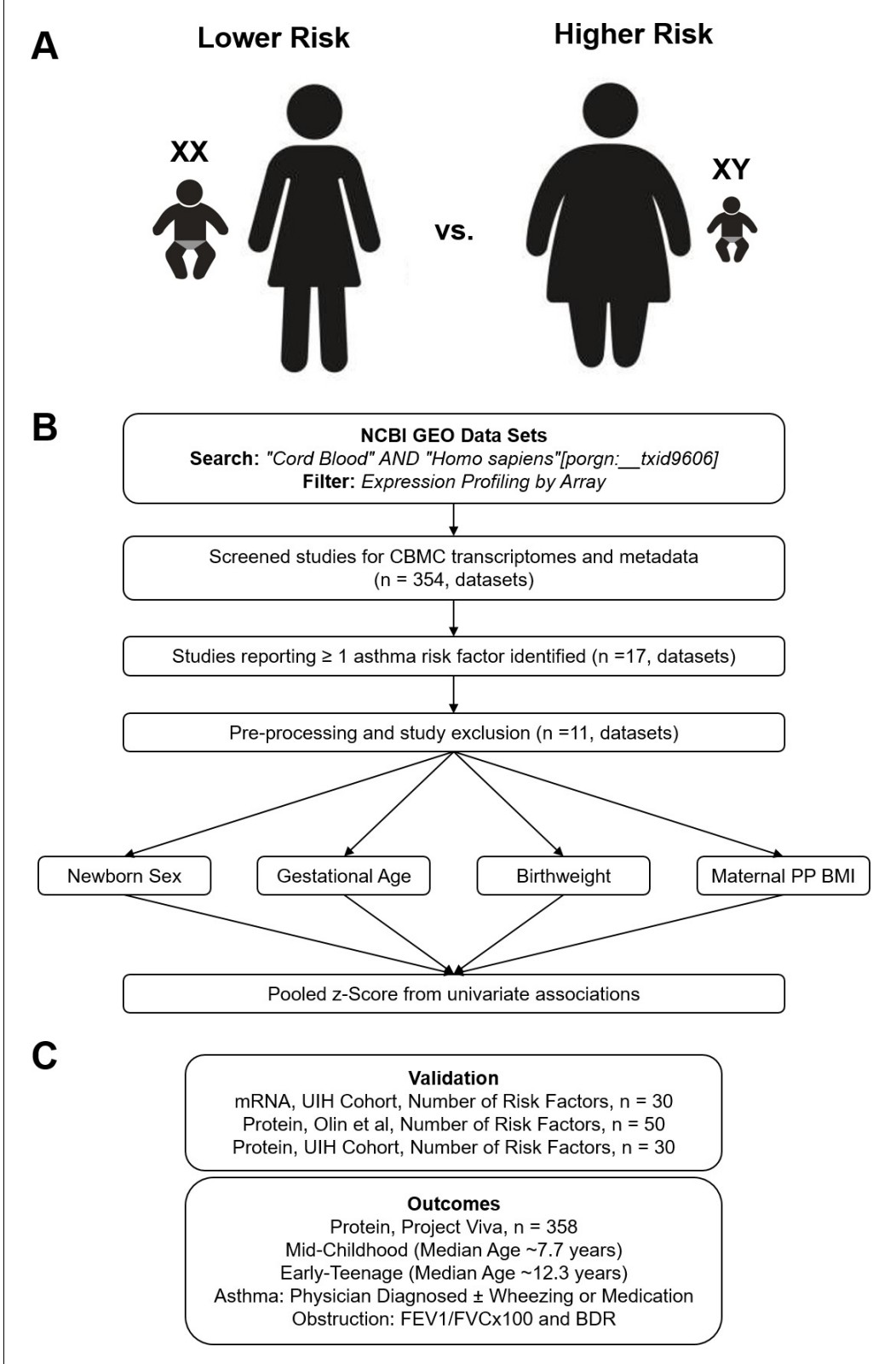

**Figure 1.** Overview of analytic approach used to identify biological risk for pediatric asthma. (**A**) Previously described perinatal risk factors for development of pediatric asthma: preterm birth, low birthweight, male, and maternal obesity. (**B**) Flow diagram of search, inclusion, exclusion, and univariate testing for transcriptomic analysis. (**C**) Cohorts, types of biosamples, and outcomes used for validation.

the identified datasets reported metadata about maternal atopy or child's ethnicity. Therefore, we chose to focus the analysis on gene expression associated with newborn sex, gestational age at birth, newborn weight, and maternal PP BMI. Six datasets were excluded due to homogenous metadata (e.g., all female). Datasets and the corresponding analysis are reported; a total of 605 unique transcriptomes were included (*Table 1*; *Bukowski et al., 2017*; *Edlow et al., 2016*; *Kallionpää et al., 2014*; *Mason et al., 2010*; *Rager et al., 2014*; *Smith et al., 2014*; *Stünkel et al., 2012*; *Turan et al., 2012*; *Votavova et al., 2011*; *Votavova et al., 2012*; *Winckelmans et al., 2017*).

To validate the findings of the meta-analysis, we assessed three independent cohorts to confirm which genes are associated with asthma risk (*Figure 1C*). Further details regarding validation and outcomes are discussed in the Results section. In brief, the goal of validation was to assess gene expression in an independent cohort (UIH Cohort [*Koenig et al., 2020*]) where all subjects had complete metadata regarding newborn sex, gestational age, birthweight, and PP BMI. We sought to further understand whether transcriptomic differences in CBMCs corresponded to differences at the protein level (all three cohorts) or cell population level (*Olin et al., 2018*). Finally, we assessed two identified proteins in another independent cohort (Project Viva Cohort [*Oken et al., 2015*]) to test for association with pediatric asthma and pulmonary function outcomes at two follow-up time points.

## Meta-analysis of CBMC transcription associated with individual perinatal risk factors

Univariate random-effects models were generated to assess transcriptional changes in CBMCs with regards to newborn sex, gestational age, birthweight, and maternal PP BMI (*Figure 2A*). Differential expression (false discovery rate [FDR] < 1%) was observed in 122, 34, 4, and 12 genes when comparing fetal sex, gestational age, birthweight, and maternal pre-pregnancy BMI, respectively (*Supplementary file 1–4*). When evaluating sex and gestational age, several expected genes were identified to have large transcriptional changes. With regards to sex-associated transcriptional changes, although there was no X or Y chromosome-wide gene enrichment, several genes located

**Table 1.** GSE data sets used for meta-analyses.

| GSE | GPL | N | Newborn sex | Gestational age | Birthweight | Maternal pre-pregnancy BMI | Title |
|-----|-----|---|-------------|-----------------|-------------|----------------------------|-------|
| GSE21342 | GPL6947 | 37 | | | | + | Maternal influences on the transmission of leukocyte gene expression profiles in population samples |
| GSE25504 | GPL570 | 20 | + | | | | Whole blood mRNA expression profiling of host molecular networks in neonatal sepsis |
| GSE27272 | GPL6883 | 64 | | + | + | + | Comprehensive study of tobacco smoke-related transcriptome alterations in maternal and fetal cells |
| GSE30032 | GPL6883 | 47 | | + | + | + | Deregulation of gene expression induced by environmental tobacco smoke exposure in pregnancy |
| GSE36828 | GPL6947 | 48 | | + | + | | Genome-wide analysis of gene expression levels in placenta and cord blood samples from newborns babies |
| GSE37100 | GPL14550 | 38 | + | + | + | | Transcriptome changes affecting hedgehog and cytokine signaling in the umbilical cord in late pregnancy: implications for disease risk |
| GSE48354 | GPL16686 | 38 | + | + | + | | Prenatal arsenic exposure and the epigenome: altered gene expression profiles in newborn cord blood |
| GSE53473 | GPL13667 | 128 | + | + | | | Standard of hygiene and immune adaptation in newborn infants |
| GSE60403 | GPL570 | 16 | + | | | + | The obese fetal transcriptome |
| GSE73685 | GPL6244 | 23 | | + | | | Unique inflammatory transcriptome profiles at the maternal fetal interface and onset of human preterm and term birth |
| GSE83393 | GPL17077 | 146 | + | | | | Newborn sex-specific transcriptome signatures and gestational exposure to fine particles: findings from the ENVIRONAGE Birth Cohort |
| | N | 605 | 386 | 386 | 235 | 164 | |

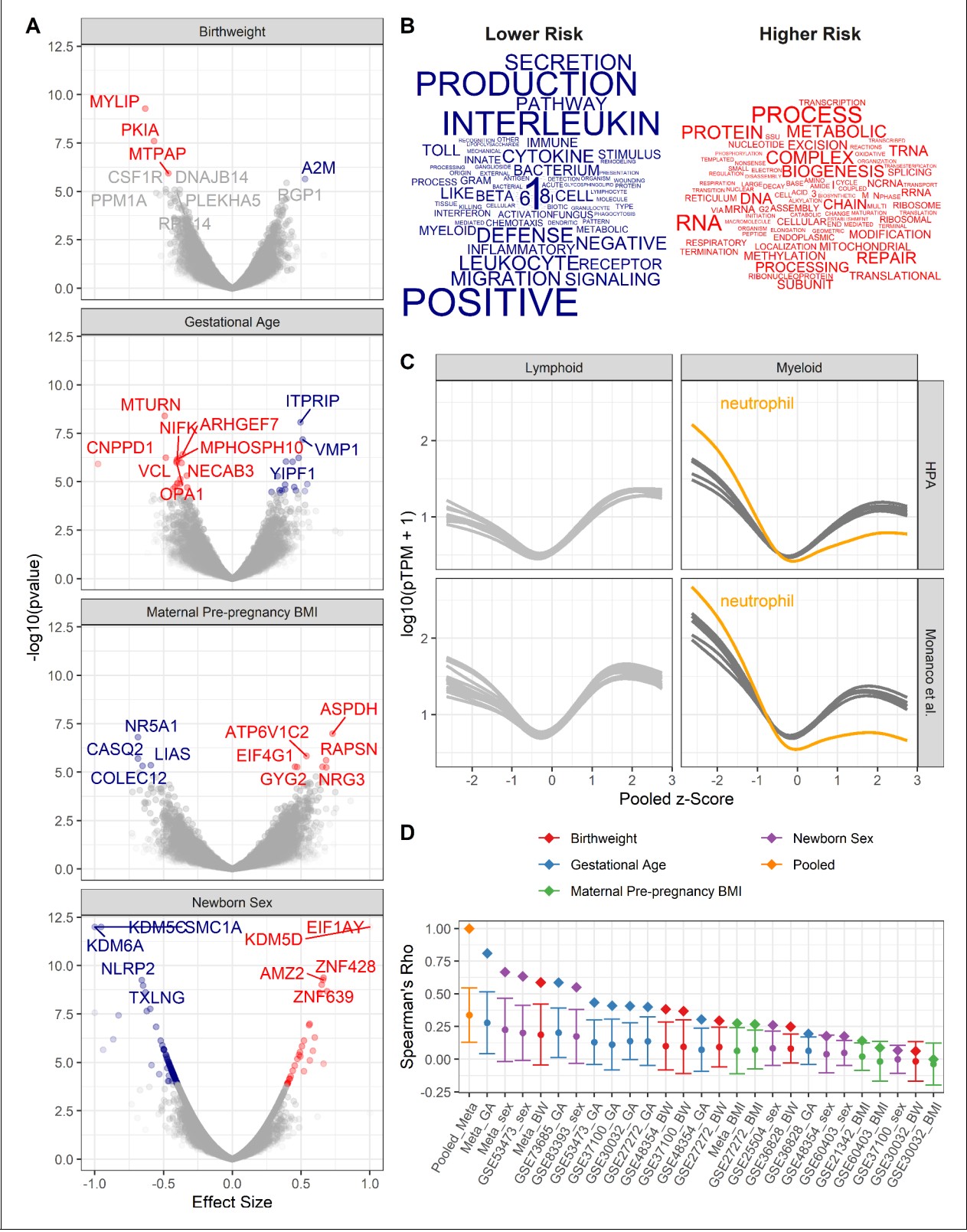

**Figure 2.** Pooled meta-analysis z-scores identify gene expression signatures related to asthma risk. Significant (FDR < 1%) genes and gene sets are colored by their association with either higher (red) or lower (blue) risk. (**A**) Volcano plots of gene expression for univariate analyses. Top 10 most significant genes labeled. (**B**) Word clouds of GO terms significantly enriched (FDR < 1%) using the pooled z-score as pre-ranked list for GSEA. (**C**) Protein coding transcripts per million reads (pTPM) in peripheral blood cells (Human Protein Atlas and Monaco et al (***Uhlen et al., 2010***;

*Figure 2 continued on next page*

*Figure 2 continued*

*Monaco et al., 2019*) relative to pooled z-score. Each line represents one cell type; neutrophils highlighted in orange. (D) Spearman's correlation between pooled z-statistic and individual analyses (diamonds). Average Spearman's correlations between individual analyses and combination of all other analyses (circle), SD indicated by error bars.

The online version of this article includes the following figure supplement(s) for figure 2:

**Figure supplement 1.** Association between differentially methylated genes and gene expression changes with gestational age.

on X (*KDM5C*, *SMC1A*, *TXLNG*, and *KDM6A*) and Y (*KDM5D* and *EIF1AY*) chromosomes exhibited the largest effect sizes and most significant differences. In addition to expected sex-associated transcriptional changes, *HBE1*, a hemoglobin subunit associated with fetal erythropoiesis, was significantly associated with preterm gestational ages. To further validate our gestational age findings, we compared our univariate analysis with previously published results of differentially methylated regions associated with gestational age at birth estimated by last menstrual period (*Bohlin et al., 2016*; *Figure 2—figure supplement 1*). A significant association was observed between differentially methylated genes and the effect size in our gestational age analysis, such that genes whose methylation increased with gestational age as reported by *Bohlin et al., 2016* showed on average decreased expression with gestational age in our meta-analysis.

## Pooled meta-analysis gene expression signature

To identify the biological processes that are enriched by genes with transcriptional changes associated with higher or lower risk of asthma, the z-scores from each univariate meta-analysis were averaged, such that negative z-statistics were associated with lower risk (female, older gestational ages, higher birthweights, and lower maternal PP BMI) and positive z-statistics were associated with increased risk. Thus, the pooled z-score indicates the average probability that a gene's expression is associated with either increased or decreased risk of asthma based on an individual's demographics and birth characteristics. The averaged z-statistic was used as a pre-ranked list for gene set enrichment analysis. GO terms were assessed for enrichment; 18 and 19 GO terms were significantly enriched (FDR < 1%) with regards to low- and high-risk profiles, respectively (*Figure 2B*). Genes associated with lower risk exhibited increased representation in GO terms involving innate immune signaling and defense, whereas high-risk genes were enriched in pathways involving translation and RNA metabolic processes.

Gene expression studies from pooled cellular populations (e.g. CBMCs, peripheral blood mononuclear cells, and tissues) can be influenced by the cellular composition. To determine whether specific cell population enrichment was associated with the pooled z-score, the Human Protein Atlas (HPA; *Uhlen et al., 2010*, *Monaco et al., 2019*) was utilized to assess the abundance of transcripts in peripheral blood leukocyte RNA transcriptomes in relationship to the pooled z-score. We observed a generalized increase in expression of low-risk genes in myeloid cells and high-risk genes in lymphoid cells. This pattern of expression was most pronounced in neutrophils (*Figure 2C*). These results suggest that lower risk individuals have increased populations of myeloid cells in their CBMCs.

A potential confounder in pooling results is the potential over-representation of any one analysis. To assess bias in the pooled z-score, two analyses were performed (*Figure 2D*). In the first assessment, z-scores from each individual analysis correlated with the pooled z-score. The highest correlations with the pooled z-score were the z-scores from the meta-analyses assessing gestational age at birth, newborn sex, and birthweight. Individual dataset z-scores for each dataset demonstrated a similar trend. In the second assessment, z-scores from each individual analysis were correlated with the combination of all other z-scores. Again, the pooled z-score had the highest average correlation followed by the meta-analyses. Together, this demonstrates that the pooled z-score does, indeed, amalgamate information across all of the analyses, with the most influence arising from gestational age at birth, newborn sex, and birthweight.

## Specific granule gene expression association with multiple pediatric asthma risk factors

To confirm gene expression changes associated with asthma risk stratification, pooled z-scores from the meta-analysis were compared with the UIH cohort, a cohort of individuals in which newborn sex, maternal PP BMI, gestational age at birth, and birthweight were known (*Supplementary file 5*). UIH cohort z-scores were calculated from mRNAseq of CBMCs, where gene expression was modeled as a function of number of risk factors (*Supplementary file 6*). We developed a method to validate the congruence between the UIH cohort and the pooled meta-analysis, which we termed the replication score (RS). This RS is the product of the pooled z-score from the meta-analysis and the UIH z-score (see Materials and methods). We assessed the relationship between RS cutoff, p-values, and number of genes (*Figure 3—figure supplement 1*). Genes with a RS greater than three were identified as being sufficiently congruent. Fifty-one genes, 0.4% of all genes, tested had a RS greater than 3 (*Figure 3A*). These identified genes corresponded well with the results of pooled z-score for gestational age at birth, newborn sex and birthweight, but showed limited correlation to maternal PP BMI. They had median p-values of 0.02, 0.01, 0.11, and 0.58 in the gestational age at birth, newborn sex, birthweight, and maternal PP BMI meta-analyses, and median p-value of 0.02 in the UIH cohort.

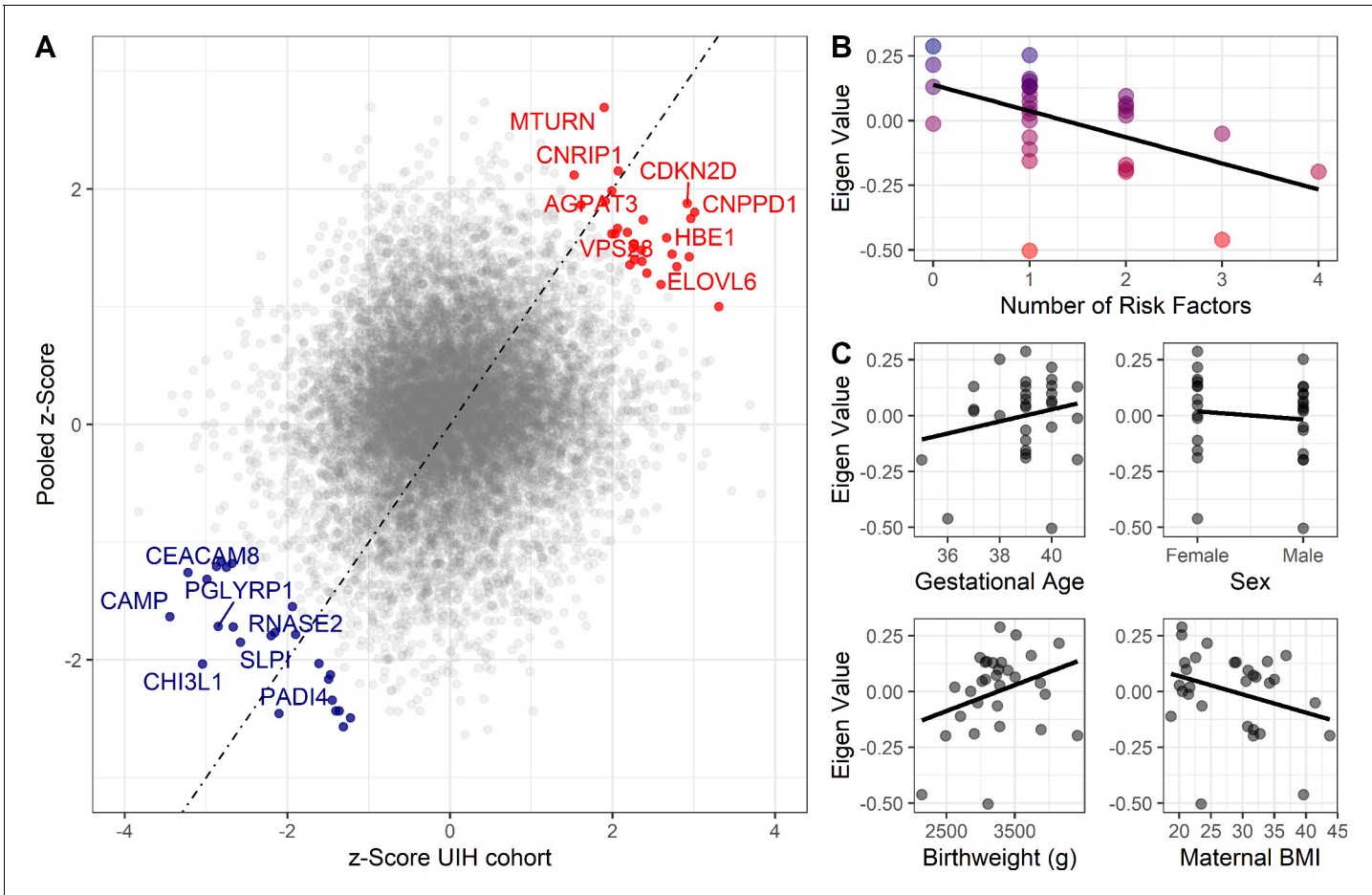

**Figure 3.** Validation cohort identifies gene signature associated with pediatric asthma risk factors. Color labeling indicating association with either higher (red) or lower (blue) risk of pediatric asthma development. (**A**) Dot-plot demonstrating validation between meta-analysis pooled z-score and UIH cohort mRNAseq z-score. Colored and labeled dots indicate those with non-parametric replication score greater than 3 and 4, respectively. (**B,C**) Association between number of risk factors or individual risk factors and eigenvalue of gene signature (validation score > 3), UIH cohort.

The online version of this article includes the following figure supplement(s) for figure 3:

**Figure supplement 1.** Replication score enriches for genes associated with multiple risk factors.

**Figure supplement 2.** Protein–protein Interaction network of candidate genes.

Replicating genes were enriched for processes involved in vesicle biology (*Figure 3—figure supplement 2*). Specifically, replicating genes associated with low risk were enriched for genes that are components of granulocyte specific granules (*MS4A3, CEACAM8, OLR1, CAMP, LTF, CHI3L1, SLPI, PGLYRP1*). With regards to genes associated with higher risk, genes involved in vesicle sorting/production (*VPS28, VTI1B, FIS1*) as well as several genes involved in vesicle membrane biology (*AGPAT3, ELOVL6, TM7SF2*) were identified.

To test whether replicating genes were associated with the number of risk factors, these 51 genes were assessed using principal component analysis. Using the first Eigenvector (explaining 41.3% of variance in replicating genes), a significant association (R [95% confidence interval (CI)=−0.51 [−0.73, –0.18], p-value<0.01) was observed between UIH cohort Eigenvalues and number of epidemiologic risk factors (*Figure 3B*). Positive eigenvalues represent increased expression of low-risk genes and negative eigenvalues represent increased expression of high-risk genes. Although the associations with individual risk factors were in the expected directions, they were not significant (*Figure 3C*), suggesting that the additive effect is greater than the individual.

## Cellular and protein abundance in relation to pediatric asthma risk factors

To further determine whether these changes are due to differences in cellular populations, we analyzed mass cytometry data published by Olin et al. for abundance of 21 different cell types in relationship to number of epidemiologic risk factors for pediatric asthma (*Votavova et al., 2011*). Cord blood neutrophil abundance was inversely associated (R [95% CI]=−0.57 [–0.73, –0.34], Bonferroni p-adj<0.001) with the number of risk factors (*Figure 4A*). Other myeloid cell types, CD14+ monocytes, and myeloid-derived dendritic cells also had negative correlations but were weaker and not significant after multiple testing correction (**data not shown**).

Extending these findings from gene expression to protein abundance, reported umbilical cord plasma-protein abundance data was correlated with number of risk factors in a secondary analysis (*Olin et al., 2018*). Serum proteins from genes that replicated with low risk had on average negative correlations with number of risk factors (*Figure 4B*). No proteins from the high-risk genes were tested in plasma, due to their intracellular localization. Notably, proteins (CEACAM8, PGLYRP-1, CHIT1, sIL6Rα, MMP-9, and OSM) predicted to be enriched in neutrophils by the HPA (*Uhlen et al., 2010*) had strong correlations with both neutrophil abundance and number of risk factors (*Figure 4B,C*).

We hypothesized that the serum concentration of proteins identified as low risk in our transcriptomic analysis would correlate with mRNA abundance in CBMCs, whereas those not associated with risk would not correlate with mRNA in CBMCs. To test this hypothesis, we used the UIH cohort to correlate mRNA abundance with serum protein concentration of PGLYRP-1 (low risk) and sIL6Rα (no risk). We observed a significant (R [95% CI]=0.39 [0.03, 0.66], p<0.05) association between PGLYRP-1 protein concentration and mRNA (*Figure 4D*). Consistent with transcriptomic results, PGLYRP-1 cord blood serum concentration was inversely associated with number of risk factors (R [95% CI] =−0.51 [–0.74, –0.17], p<0.01). sIL6Rα was neither associated with its mRNA in CBMCs (R [95% CI] =0.37 [–0.22, 0.77], p=0.21) nor associated with the number of risk factors (R [95% CI]=0.20 [–0.39, 0.67], p=0.50).

## Demographic associations with serum neutrophil proteins in UIH and Project Viva cohorts

We tested the association between PGLYRP-1 and sIL6Rα, individual risk factors, and demographics in the UIH and Project Viva cohorts. Cord blood serum was available in a subset of individuals (n = 358) from Project Viva (*Supplementary file 7*). There was no significant difference (p>0.05, Wilcoxon rank sum test) in PGLYRP-1 or sIL6Rα between UIH and Project Viva cohorts (*Figure 4—figure supplement 1*). Consistent with our previous observations, PGLYRP-1 and sIL6Rα were positively correlated in both UIH (R [95% CI]=0.21 [−0.16, 0.54]) and Project Viva (R [95% CI]=0.19 [0.09, 0.29]) cohorts. Similar to the observation in the UIH cohort, there was a negative association between PGLYRP-1 and number of risk factors in Project Viva (β [95% CI]=−0.22 [−0.36, –0.07], p-value=0.003) (*Table 2*). This association was driven by the relationship between PGLYRP-1, gestational age, and sex. There was no association with PP BMI or birthweight when taking into account

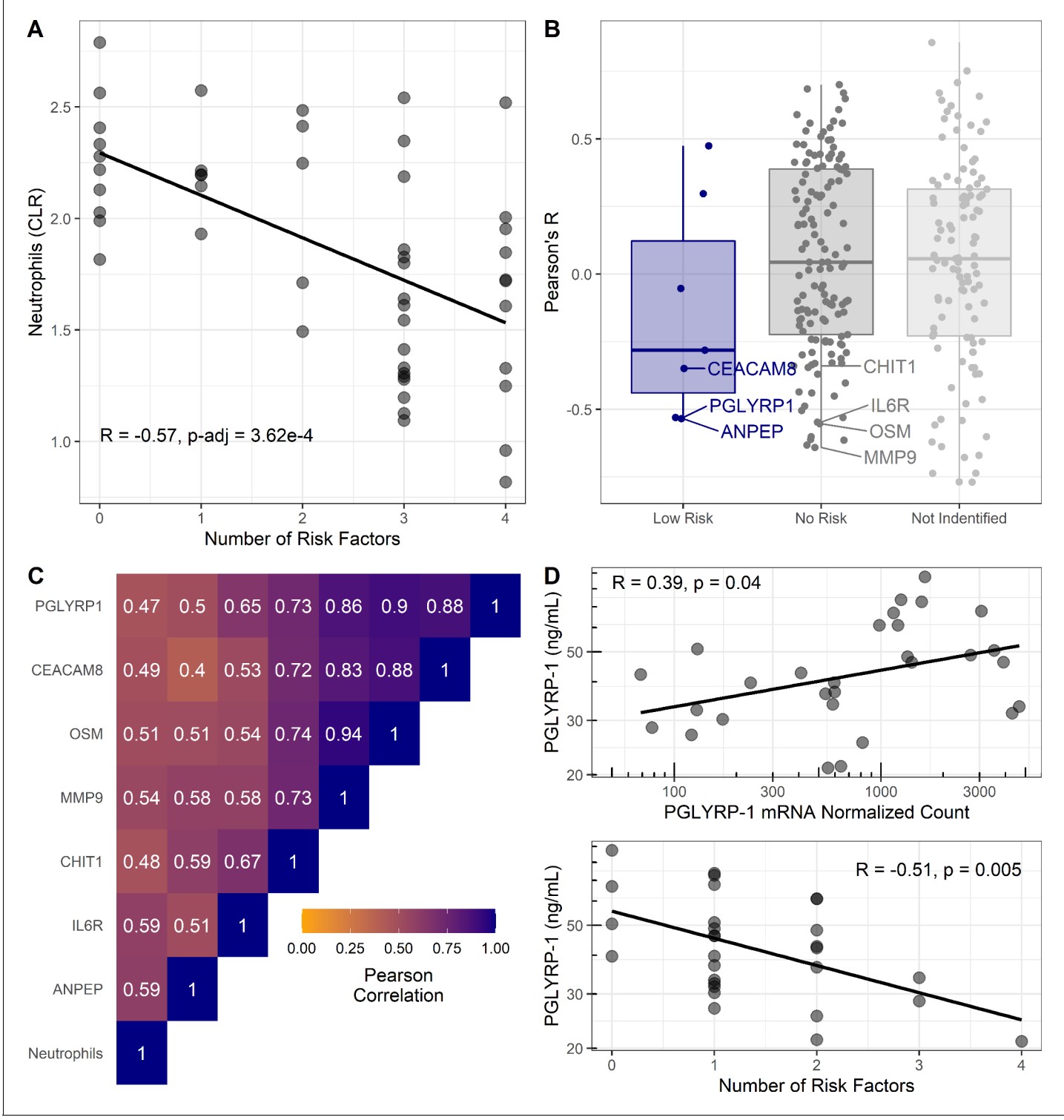

**Figure 4.** Cellular and proteomic differences associated with pediatric asthma risk factors. (A–C) Re-analysis of publicly available data from *Olin et al., 2018*. (A) Percentage of neutrophils in cord blood (transformed using centered log-ratios, CLR) correlated with number of risk factors. Pearson's correlation (R) and Bonferroni adjusted p-value reported. (B) Pearson's correlation coefficients (R) for plasma-protein concentration and number of risk factors distributed based on risk association of proteins as per *Figure 3*. Corresponding mRNA from CBMCs were identified for low-risk associated proteins (blue) and no risk associated proteins (dark gray). Most significant negative protein correlations with neutrophil-enriched mRNA (Human Protein Atlas [*Uhlen et al., 2010*]) are notated. Proteins identified in previous analysis without corresponding mRNA shown light gray. (C) Heatmap of

*Figure 4 continued on next page*

Figure 4 continued

Pearson's correlations between neutrophils and neutrophil-derived proteins identified in (B). (D) Association between PGLYRP-1 umbilical cord serum concentration, PGLYRP-1 CBMC mRNA, and number of risk factors in UIH cohort.

The online version of this article includes the following figure supplement(s) for figure 4:

**Figure supplement 1.** Association between PGLYRP-1 and sIL6Rα in UIH and Project Viva cohorts.

gestational age and sex. Furthermore, there was no association of sIL6Rα with the number or risk factors or any individual risk factor. Interestingly, there was an inverse relationship observed in both the UIH and Project Viva cohorts between sIL6Rα and self-reported maternal race as Black/African-American. Collectively, these results suggest that increased abundance of mRNA from genes localizing to neutrophil-specific granules are associated with the number of risk factors for pediatric asthma. These changes in mRNA are reflected in the abundance of these specific granule proteins in serum and plasma.

## Serum neutrophil protein association with pediatric pulmonary outcomes

In context of our previous results, we hypothesized that specific granule protein abundance in serum is associated with risk of pediatric asthma and this process is independent of neutrophil abundance. To evaluate this hypothesis, we measured PGLYRP-1 (present in neutrophil-specific granule and correlates with its mRNA in CBMCs) and sIL6Rα (derived from neutrophils but not present in specific granules and does not correlate with its mRNA in CBMCs) in umbilical cord blood serum. At two follow-up time points, asthma outcomes and expiratory flow volumes were modeled as a function of PGLYRP-1 and sIL6Rα in a subset of individuals in Project Viva (*Figure 5—figure supplement 1*). The demographics of the subset of individuals from Project Viva from which umbilical cord serum was available had a similar demographic profile as the full cohort. One notable difference was a sizeable decreased response rate for asthma outcomes at the early-teenage follow-up compared to the full cohort (32% subset vs. 47% full cohort).

PGLYRP-1 and sIL6Rα were modeled as predictors for current asthma at mid-childhood (median age ~7.7 years old) and early-teen (median age ~12.3 years old) follow ups (*Table 3*). Four regression models were used to estimate the association between asthma outcomes: univariate, adjustment for child's birth characteristics and demographics, adjustment for mother's demographics, and adjustment for birth characteristics and all demographics (reported in manuscript). The abundance of PGLYRP-1 was significantly associated with current asthma at mid-childhood (adjusted odds ratio [OR] [95% CI]: 0.50 [0.31, 0.77] per 1 SD increase, p-value=0.003) (*Figure 5A*). There were no significant associations between current asthma and PGLYRP-1 at the early-teen follow up; however, the CI at this time point was much wider, likely secondary to the smaller sample size. There were no significant associations between sIL6Rα with any asthma outcome at either time point.

We also performed analyses to estimate the relationship of cord blood PGLYRP-1 and sIL6Rα with $FEV_1/FVCx100$ ratio and bronchodilator response (BDR) at mid-childhood and early-teen follow-up time points (*Table 4*). There was no significant association between sIL6Rα and $FEV_1/FVCx100$ ratio at either time point. PGLYRP-1 and sIL6Rα were not associated with BDR at either time point. However, there was trend toward an association between PGLYRP-1 concentration and $FEV_1/FVCx100$ ratio at mid-childhood (adjusted β [95% CI]: 1.18 [–0.18, 2.56] per 1 SD increase, p-value=0.09) and significant association at the early-teen follow up (adjusted β [95% CI]: 1.15 [0.20, 2.10] per 1 SD increase, p-value=0.02) (*Figure 5B*).

To further our understanding of the relationship between PGLYRP-1 and outcomes, we performed two secondary analyses of models adjusted for all demographics and birth characteristics. First, we assessed the total variance explained by the regression model adjusting for both birth characteristics and demographics, and the variance explained by each of the individual predictors in the model. Assessing current asthma risk at mid-childhood, the regression model explained approximately 18% of the variance, and PGLYRP-1 was the most important predictor. Assessing current $FEV_1/FVCx100$ at early-teen time point, the model explained approximately 26% of variance, and

**Table 2.** Univariate associations between demographics and serum proteins.

| | UIH (n = 29) | | Project viva (n = 358) | |
|---|---|---|---|---|
| | β (95% CI) | p-value | β (95% CI) | p-value |
| **PGLYRP-1 Z-score*** | | | | |
| Number of risk factors[†] | −0.54 (−0.88, −0.19) | 0.005 | −0.22 (−0.36, −0.07) | 0.003 |
| Maternal race: White (ref) | 0 (ref) | | 0 (ref) | |
| Maternal race: Black | 0.01 (−0.83, 0.86) | 0.97 | 0.15 (−0.16, 0.46) | 0.34 |
| Maternal race: Hispanic | 1.03 (0.05, 2.01) | 0.04 | 0.40 (−0.07, 0.86) | 0.10 |
| Maternal race: Other | 1.54 (−0.46, 3.54) | 0.12 | 0.10 (−0.25, 0.46) | 0.57 |
| Maternal atopy | −0.14 (−0.92, 0.64) | 0.72 | 0.06 (−0.16, 0.28) | 0.58 |
| Maternal pre-pregnancy BMI | −0.03 (−0.09, 0.02) | 0.25 | 0.01 (−0.01, 0.03) | 0.56 |
| Maternal smoking: never (ref) | 0 (ref) | | 0 (ref) | |
| Maternal smoking: former | 0.09 (−0.92, 1.12) | 0.84 | −0.16 (−0.44, 0.11) | 0.25 |
| Maternal smoking: during pregnancy | – | – | −0.12 (−0.47, 0.22) | 0.48 |
| Maternal college graduate | 0.07 (−0.80, 0.93) | 0.87 | −0.12 (−0.34, 0.10) | 0.27 |
| Any antibiotic use during pregnancy | – | – | 0.20 (-0.02, 0.43) | 0.08 |
| Gestational age weeks | 0.25 (−0.01, 0.51) | 0.06 | 0.12 (0.06, 0.19) | 0.0003 |
| Birthweight adj GA and Sex (Z-score) | 0.33 (−1.07, 1.74) | 0.63 | 0.03 (−0.09, 0.14) | 0.66 |
| Female | 0.39 (−0.37, 1.15) | 0.30 | 0.31 (0.11, 0.52) | 0.003 |
| C-section | −0.08 (−0.89, 0.74) | 0.85 | −0.29 (−0.55, −0.02) | 0.03 |
| Child's race: White (ref) | – | – | 0 (ref) | |
| Child's race: Black | – | – | 0.16 (−0.14, 0.46) | 0.31 |
| Child's race: Hispanic | – | – | 0.18 (−0.31, 0.68) | 0.46 |
| Child's race: Other | – | – | 0.06 (-0.27, 0.38) | 0.73 |
| **sIL6Rα Z-score*** | | | | |
| Number of risk factors[†] | −0.14 (−0.54, −0.25) | 0.48 | 0.02 (−0.13, 0.16) | 0.81 |
| Maternal race: White (ref) | 0 (ref) | | 0 (ref) | |
| Maternal race: Black | −1.04 (−1.90, −0.19) | 0.02 | −0.29 (−0.60, 0.02) | 0.07 |
| Maternal race: Hispanic | −0.36 (−1.34, 0.62) | 0.45 | 0.34 (-0.12, 0.80) | 0.15 |
| Maternal race: Other | 0.4 (−1.61, 2.42) | 0.68 | −0.03 (−0.39, 0.32) | 0.85 |
| Maternal atopy | −0.23 (−1.01, 0.54) | 0.55 | −0.08 (−0.30, 0.14) | 0.47 |
| Maternal pre-pregnancy BMI | −0.04 (−0.10, 0.01) | 0.12 | 0.00 (−0.02, 0.02) | 0.72 |
| Maternal smoking: never (ref) | 0 (ref) | | 0 (ref) | |
| Maternal smoking: former | −0.90 (−1.86, 0.06) | 0.07 | −0.23 (−0.50, 0.05) | 0.10 |
| Maternal smoking: during pregnancy | – | – | −0.12 (−0.46, 0.22) | 0.48 |
| Maternal college graduate | 0.44 (−0.41, 1.29) | 0.29 | −0.02 (−0.24, 0.19) | 0.84 |
| Any antibiotic use during pregnancy | – | – | 0.04 (−0.19, 0.26) | 0.76 |
| Gestational age weeks | −0.11 (−0.39, 0.17) | 0.43 | −0.04 (−0.10, 0.03) | 0.29 |
| Birthweight adj GA and sex (Z-score) | −0.15 (−1.57, 1.26) | 0.82 | −0.07 (−0.18, 0.04) | 0.23 |
| Female | 0.48 (−0.28, 1.23) | 0.21 | 0.05 (−0.16, 0.26) | 0.63 |
| C-section | −0.60 (−1.38, 0.18) | 0.12 | −0.14 (−0.41, 0.12) | 0.29 |
| Child's race: White (ref) | – | – | 0 (ref) | |
| Child's race: Black | – | – | −0.24 (−0.54, 0.05) | 0.11 |
| Child's race: Hispanic | – | – | 0.27 (−0.22, 0.76) | 0.28 |
| Child's race: Other | – | – | 0.02 (−0.30, 0.34) | 0.90 |

*Serum protein concentrations for UIH and Project Viva were log10 transformed and converted into an internal Z-score.

[†]Number of risk factors determined by preterm birth, maternal BMI > 29.9, male, birthweight (z-score) < −1.

**Table 3.** Association between serum protein concentration and asthma outcomes.

| | | Mid-childhood | | Early-teen | |
|---|---|---|---|---|---|
| | | Current asthma, OR (95% CI) | Ever asthma, OR (95% CI) | Current asthma, OR (95% CI) | Ever asthma, OR (95% CI) |
| PGLYRP-1 Z-score* | | | | | |
| | Univariate | 0.52 (0.35, 0.75) | 0.52 (0.36, 0.74) | 0.65 (0.39, 1.10) | 0.64 (0.45, 0.89) |
| | Model 1[†] | 0.57 (0.37, 0.85) | 0.54 (0.36, 0.79) | 0.86 (0.48, 1.54) | 0.72 (0.50, 1.03) |
| | Model 2[‡] | 0.57 (0.26, 0.65) | 0.41 (0.26, 0.61) | 0.62 (0.35, 1.09) | 0.61 (0.42, 0.87) |
| | Model 3[#] | 0.50 (0.31, 0.77) | 0.48 (0.31, 0.77) | 0.88 (0.45, 1.72) | 0.74 (0.51, 1.07) |
| sIL6Rα Z-score* | | | | | |
| | Univariate | 0.87 (0.61, 1.23) | 0.93 (0.67, 1.29) | 0.75 (0.42, 1.28) | 0.93 (0.66, 1.29) |
| | Model 1[†] | 0.83 (0.56, 1.23) | 0.89 (0.63, 1.30) | 0.68 (0.37, 1.21) | 0.90 (0.63, 1.27) |
| | Model 2[‡] | 0.83 (0.57, 1.24) | 0.90 (0.63, 1.28) | 0.67 (0.33, 1.27) | 0.93 (0.65, 1.31) |
| | Model 3[#] | 0.83 (0.55, 1.25) | 0.90 (0.62, 1.30) | 0.60 (0.27, 1.19) | 0.88 (0.61, 1.27) |

*Serum protein concentrations were log10 transformed and converted into an internal Z-score.

[†]Serum protein concentrations were log10 transformed and conver gestational age, birthweight adjusted for gestational age, mode of delivery, child's sex, child's race/ethnicity.

[‡](Mother's demographics): adjusted for maternal pre-pregnancy BMI, maternal race/ethnicity, maternal level of education, maternal atopy, antibiotic exposure during pregnancy, smoking during pregnancy, 6 months or 1 year.

[#]Model 3 (all demographics and birth characteristics): adjusted for all demographics and characteristics in models 1 and 2 except maternal race/ethnicity. This reported value in manuscript.

PGLYRP-1 was the second most important predictor (*Figure 5—figure supplement 2*). Second, to identify covariates that modify the effect of PGLYRP-1, we performed subset analyses (*Figure 5—figure supplement 3*). Small for gestational age and children identified by their mothers as 'Other' race/ethnicity displayed significantly different associations between PGLYRP-1 and mid-childhood asthma. Small for gestational age and children with obese mothers displayed significantly different associations between PGLYRP-1 and FEV$_1$/FVC.

## Discussion

Our study has identified a novel association between epidemiologic risk, neutrophil-specific granules, and pediatric pulmonary outcomes. These findings implicate a role for PGLYRP-1 and other specific granule proteins as predictors of pediatric asthma risk and pulmonary function. This is in contrast to sIL6Rα, which is not localized to specific granules, its protein abundance is not regulated by transcription, and is not associated with any pulmonary outcomes.

By pooling the meta-analysis results, we established an association between multiple risk factors and the expression of genes involved in innate immunity and nucleic acid metabolism. We hypothesized that this gene signature represents increased myelopoiesis in utero and correlates with perinatal risk for pediatric asthma. During the process of myeloid cell differentiation, production of proteins responsible for defense against microbes and pro-inflammatory signaling (e.g., IL-1β) are amplified, while translational activity and nucleolar size wane (*Zhu et al., 2017*; *Grassi et al., 2018*). In our analysis, lower risk genes had higher expression in myeloid cells, most notably neutrophils. The low-risk genes were enriched for those that are implicated in defense responses towards bacteria and fungi. Additionally, this gene signature was strongly correlated with gestational age at birth, newborn sex, and birthweight (weaker association with maternal PP BMI). Our findings parallel previous literature, which has demonstrated that preterm birth, male sex, and low birthweight are associated with reduced abundance of neutrophils and monocytes (*Glasser et al., 2015*).

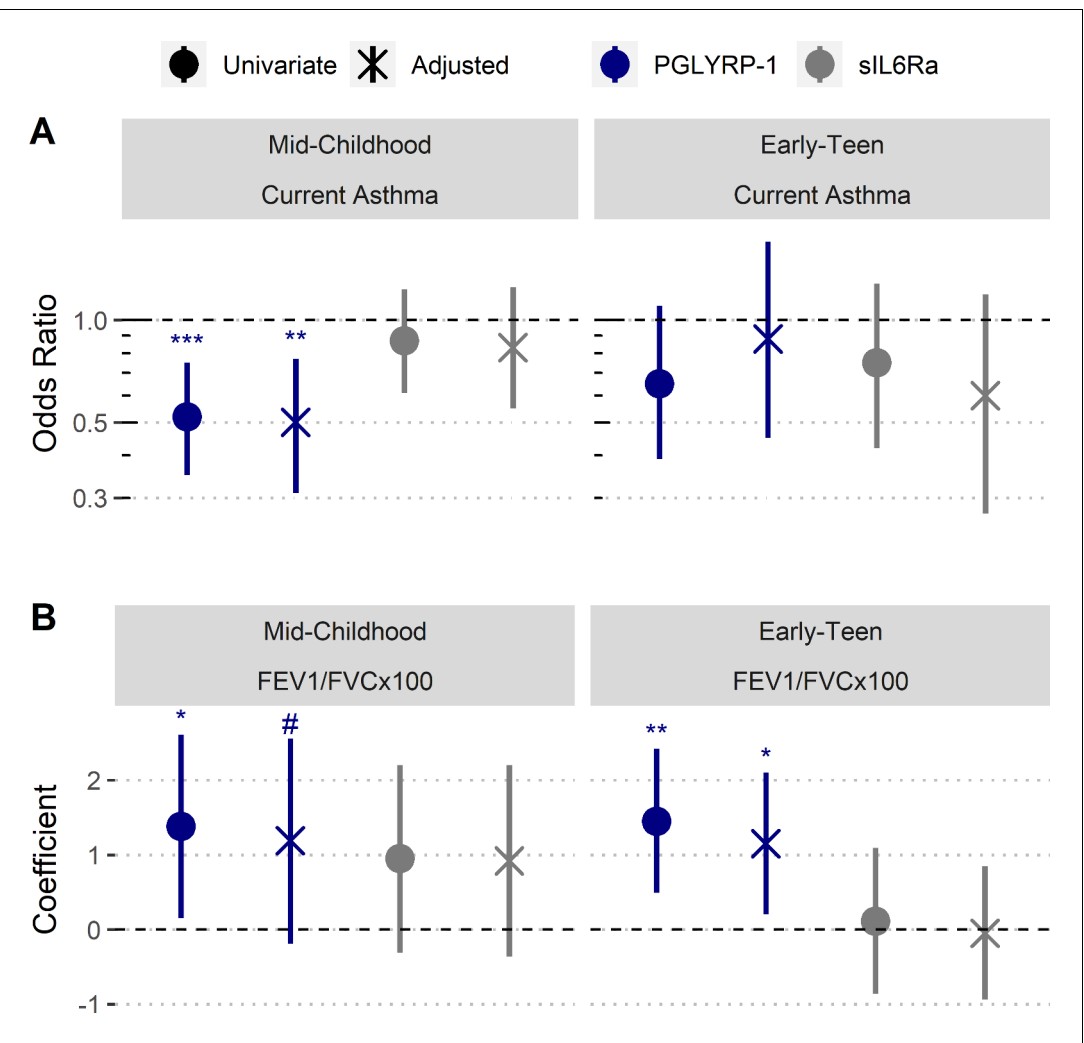

**Figure 5.** Increased umbilical cord blood serum PGLYRP-1 is associated with increased FEV$_1$/FVC and reduced odds of pediatric asthma. Samples and data derived from a subset of Project Viva (n=358). Odds ratio and coefficient estimates are based on 1 SD increase in serum proteins (PGLYRP-1, sIL6Rα). Error bars indicate 95% CI. Adjusted model co-variates: gestational age, birthweight adjusted for gestational age and sex, mode of delivery, child's sex, child's race/ethnicity, maternal pre-pregnancy BMI, maternal level of education, maternal atopy, antibiotic exposure during pregnancy, and early-life smoke exposure. (**A**) PGLYRP-1 and sIL6Rα concentrations in umbilical cord blood serum association with current asthma at mid-childhood and early-teenage time points (determined by questionnaire responses). (**B**) PGLYRP-1 and sIL6Rα concentrations in umbilical cord blood serum association with FEV$_1$/FVCx100 at mid-childhood and early-teenage follow ups. ***p<0.001, **p<0.01, *p<0.05, #p<0.1.

The online version of this article includes the following figure supplement(s) for figure 5:

**Figure supplement 1.** Cord blood serum proteins in relationship to outcomes.

**Figure supplement 2.** Relative importance of predictors for pediatric asthma and FEV$_1$/FVC.

**Figure supplement 3.** Subset analysis for all covariates used in regression models.

Our replication of the pooled meta-analysis with the UIH cohort pointed toward genes located and involved in the biology of neutrophil-specific (secondary) granules. In particular, lower risk individuals had higher expression of genes whose protein products are luminal (*PGLRYP1*, *LTF*, *PTX3*, *CHI3L1*, *CAMP*, *SLPI*) and membrane (*CEACAM8*, *MS4A3*, *OLR1*) components of specific granules (**Rørvig et al., 2013**). We demonstrate that the additive effect of multiple risk factors is associated

**Table 4.** Association between serum protein concentration and pulmonary function.

| | | Mid-childhood | | Early-teen | |
|---|---|---|---|---|---|
| | | FEV$_1$/FVCx100 β (95% CI) | BDR β (95% CI) | FEV$_1$/FVCx100 β (95% CI) | BDR β (95% CI) |
| PGLYRP-1 Z-score* | | | | | |
| | Univariate | 1.38 (0.15, 2.61) | 0.10 (−1.83, 2.03) | 1.45 (0.49, 2.42) | −0.68 (−1.54, 0.17) |
| | Model 1[†] | 1.12 (−0.18, 2.41) | 0.53 (−1.44, 2.51) | 1.05 (0.11, 1.98) | −0.49 (−1.40, 0.41) |
| | Model 2[‡] | 1.38 (0.08, 2.69) | 0.61 (−1.40, 2.63) | 1.53 (0.54, 2.53) | −0.58 (−1.49, 0.34) |
| | Model 3[#] | 1.19 (−0.19, 2.56) | 1.00 (−1.04, 3.04) | 1.15 (0.20, 2.10) | −0.35 (−1.29, 0.59) |
| sIL6Rα Z-score* | | | | | |
| | Univariate | 0.95 (−0.31, 2.20) | −0.38 (−2.45, 1.69) | 0.11 (−0.86, 1.09) | 0.65 (−0.24, 1.54) |
| | Model 1[†] | 0.96 (−0.30, 2.21) | −0.37 (-2.41, 1.67) | 0.02 (−0.87, 0.92) | 0.71 (−0.19, 1.61) |
| | Model 2[‡] | 0.88 (−0.40, 2.16) | −0.62 (−2.62, 1.39) | −0.02 (−1.00, 0.97) | 0.75 (−0.17, 1.66) |
| | Model 3[#] | 0.92 (−0.36, 2.20) | −0.11 (−0.96, 0.70) | −0.05 (−0.94, 0.85) | 0.79 (−0.12, 1.69) |

*Serum protein concentrations were log10 transformed and converted into an internal Z-score.

[†]Serum protein concentrations were log10 transformed and converted into an internal Z-scoreupplemental Data\\Table 4_Association pulmonary outcomes.xlsx' "Shed's race/ethnicity.

[‡](Mother's demographics): adjusted for maternal pre-pregnancy BMI, maternal race/ethnicity, maternal level of education, maternal atopy, antibiotic exposure during pregnancy, smoking during pregnancy, 6 months or 1 year.

[#]Model 3 (all demographics and birth characteristics): adjusted for all demographics and characteristics in models 1 and 2 except maternal race/ethnicity. This the reported value in manuscript.

with the reduction of transcription and protein products of specific granules in umbilical cord blood serum. Notably, our re-analysis of mass cytometry and proteomic data demonstrated a correlation between PGLYRP-1 in serum and neutrophil abundance (*Olin et al., 2018*). Previous literature has shown that deficiency of secretory leukocyte protease inhibitor (SLPI) leads to impairment of neutrophil development and abundance (*Klimenkova et al., 2014*). Furthermore, individuals with specific granule deficiency syndrome have abnormal neutrophil morphology, increased susceptibility to infections, and increased risk of acute myeloid leukemia (*Lekstrom-Himes et al., 1999*). Thus, this further supports the notion that neutrophil differentiation and survival is partially dependent on secondary granule generation.

To further investigate how these findings are related to pediatric asthma, we chose to compare PGLYRP-1, sIL6Rα, and their associations with asthma and lung function outcomes. PGLYRP-1 and sIL6Rα were chosen because both were correlated with neutrophil abundance, yet they are derived from different processes. Variation in abundance of PGLYRP-1 in serum is due to changes in neutrophil degranulation and transcription of *PGLYRP1*, whereas sIL6Rα is derived from receptor shedding and differential splicing of its mRNAs (*Jones et al., 2001*; *Read et al., 2015*). Thus, if perinatal neutrophil abundance is associated with pediatric asthma, both PGLYRP-1 and sIL6Rα should demonstrate associations with these outcomes. In contrast, we observed a significant relationship between mid-childhood asthma and PGLYRP-1, not sIL6Rα. These data indicate that serum abundance of specific granule contents likely has a larger and more significant association with pediatric asthma risk compared to the abundance of cord blood neutrophils.

PGLYRP-1 concentration was associated with gestational age, sex, and mode of delivery. Gestational age at birth, sex, mode of delivery, and birthweight may influence not only the abundance of neutrophils, but also their functionality (e.g. phagocytosis, cytokine production, respiratory burst) (*Lawrence et al., 2017*). In contrast, sIL6Rα was only associated with maternal self-reported race, specifically in those who were Black/African-American. Notably, individuals of African descent on average have baseline lower neutrophil counts without any functional consequences (*Reich et al., 2009*). Together, this highlights that PGLYRP-1 is a potential serologic marker of neutrophil functional maturity whereas sIL6Rα likely represents the total neutrophil abundance at birth.

Whether PGLYRP-1 has a causal role in asthma pathogenesis or is merely a biomarker for risk remains to be determined. PGLYRP-1 has antimicrobial function, although the concentration we observed in cord blood is below those reported for in vitro studies (*Kashyap et al., 2011*; *Liu et al., 2000*; *Wang et al., 2007*). PGLYRP-1 functions synergistically in vitro with other antimicrobials (e.g. lysozyme), so potential benefit as an antimicrobial cannot be ruled out considering that *pglyrp-1*$^{-/-}$ mice are more susceptible to infections (*Liu et al., 2000*; *Wang et al., 2007*; *Dziarski et al., 2003*; *Ghosh et al., 2009*; *Gupta et al., 2020*; *Osanai et al., 2011*). Interestingly, mammalian PGLYRP-1 does not hydrolyze peptidoglycan, and its orthologs appear divergent from ancestral PGLYRPs, which contain enzymatic activity; thus, it may have roles other than that of an antimicrobial (*Dziarski and Gupta, 2006*). It is also possible that other individual specific granule proteins or their cumulative effects are associated with pediatric asthma. Further studies will be required to determine a mechanistic role for either PGLYRP-1 or other specific granule proteins.

Several murine studies link PGLYRP-1 to increased airway resistance and perturbed immunity in response to house dust mite (*Park et al., 2013*; *Yao et al., 2013*). Contextualization of the murine studies with our results is largely confounded by the fact these murine studies have utilized adult mice deficient in PGLYRP-1 via germ-line deletion. These studies have largely demonstrated that the genetic absence of *pglyrp1* leads to moderately reduced airway resistance, IgE, and T2 cytokine production in response to house dust mite exposure after sensitization. Notably, genetic absence of *pglyrp1* does not completely protect against airway inflammatory changes. Furthermore, one study has demonstrated that bone marrow reconstitution with WT bone marrow prior allergen sensitization abrogates this effect (*Yao et al., 2013*), suggesting PGLYRP-1 presence in adult mice at the time of sensitization is responsible for increased airway resistance and perturbed immunity. These results are not entirely surprising as PGLYRP-1 is reported to be a pro-inflammatory TREM-1 ligand and thus may propagate inflammation (*Read et al., 2015*). Extending this point, there are reported associations between increased serum PGLYRP-1 and systemic inflammatory conditions in adulthood (e.g. rheumatoid arthritis, cardiovascular disease) (*Luo et al., 2019*; *Rohatgi et al., 2009*). Beyond its role at time sensitization, PGLYRP-1 and many of the other specific granule proteins are dramatically reduced in serum concentration 1 week postnatal compared to cord blood and not well correlated with their cord blood concentration, suggesting an important temporal role and variation of these proteins (*Olin et al., 2018*). This temporal variation is largely ignored in experimental approaches such as germ-line knock outs.

Although our findings contribute to our understanding of the risk for pediatric asthma, there are several limitations. Our data is derived from observational data that limits any interpretation of causation and is potentially susceptible to confounding by unmeasured variables. The findings of association between PGLYRP-1 mid-childhood asthma and adolescent FEV$_1$/FVC were robust when adjusting for measured confounders and subset analysis. Additionally, transcription in cord blood *pglyrp1* in our meta-analysis and PGLYRP-1 concentration in the three cohorts was significantly associated with both sex and gestational age. This suggests that our results are plausibly generalizable to a larger population. Additionally, as with many prospective cohorts, the Project Viva cohort had increasing loss to follow up over time. At the early-teen follow up, we did not observe a significant association with current asthma. It is important to note that there was still an association between PGLYRP-1 and FEV$_1$/FVC at this time point. These observations could be due to two possibilities. First, at early-teen follow up, there was only 9% prevalence of current asthma in the Project Viva subset, whereas the full cohort had a 15% prevalence. This difference is likely due to higher non-response rates in the asthmatic sub-group as 24 of the 35 (68%) had missing responses, whereas only 46 of 171 (27%) of non-asthmatics had missing responses. This lead to a reduction in power at the early-teen follow-up time point, potentially leading to a false negative result. Second, PGLYRP-1 might be inversely associated with FEV$_1$/FVC in adolescence secondary to early-life pulmonary dysfunction (e.g. mid-childhood asthma). Reduced expiratory pulmonary function later in adolescences and even into adulthood is associated with individuals who were diagnosed with asthma in childhood (*Bui et al., 2018*; *Lo et al., 2020*; *Piccioni et al., 2015*; *Turturice et al., 2017b*). We view these possibilities as equally probable as current pediatric asthma can also impair FEV$_1$/FVC (*Tse et al., 2013*). Further investigation will be needed to elucidate the role of PGLYRP-1 in adolescent asthma.

In conclusion, we have identified a neutrophil development gene signature that is associated with perinatal asthma risk. A soluble specific granule protein, PGLYRP-1, was strongly associated with odds of asthma in childhood and pulmonary function in childhood and adolescence. This suggests that

perinatal granulopoiesis has a significant impact on the development of pediatric asthma and lung function.

# Materials and methods

## Key resources table

| Reagent type (species) or resource | Designation | Source or reference | Identifiers | Additional information |
|---|---|---|---|---|
| Gene (*Homo sapiens*) | *Homo sapiens* Genome Assembly | Ensembl | GRCh38.12 | |
| Biological sample (*Homo sapiens*) | Primary Cord Blood Mononuclear Cells | Volunteers | UIH Cohort | Demographics reported in **Supplementary file 5** |
| Biological sample (*Homo sapiens*) | Cord Blood Serum | Volunteers | UIH Cohort Project Viva | Demographics reported in **Supplementary files 5** and **7** |
| Commercial assay or kit | Human PGLYRP1/PGRP-S DuoSet ELISA | R and D Systems | DY2590 | |
| Commercial assay or kit | Human IL6Ra DuoSet ELISA | R and D Systems | DY227 | |
| Commercial assay or kit | RNeasy Mini Kit | Qiagen | 74104 | |
| Commercial assay or kit | RNA 6000 Nano Kit | Agilent | 5067–1511 | |
| Commercial assay or kit | Qubit RNA HS Assay Kit | Thermo Fisher Scientific | Q32852 | |
| Commercial assay or kit | TruSeq Stranded mRNA Library Prep Kit | Illumina | 20020594 | |
| Commercial assay or kit | HiSeq × Ten Reagent Kit v2.5 | Illumina | FC-501–2501 | |
| Software, algorithm | R | R | Version 3.6.3 | |
| Software, algorithm | *geoquery* | Bioconductor | Version: 2.36.0 | |
| Software, algorithm | *GeneMeta* | Bioconductor | Version: 1.54.0 | |
| Software, algorithm | *tximport* | Bioconductor | Version: 1.10.1 | |
| Software, algorithm | *DESeq2* | Bioconductor | Version: 1.22.2 | |
| Software, algorithm | *relaimpo* | CRAN | Version: 2.2–3 | |
| Software, algorithm | *salmon* | Github | Version 0.12.00 | |
| Software, algorithm | *GSEA* | gsea-msigdb.org | Version 4.0 | |

## Human subject Project Viva cohort

The current study was approved by the University of Illinois at Chicago IRB (#2016–0326) and the IRB of Harvard Pilgrim Health Care. Volunteers were recruited from women attending their first prenatal visit at one of eight practices of Atrius Harvard Vanguard Medical Associates. The exclusion criteria were multiple gestation, inability to answer questions in English, gestational age $\geq$ 22 weeks at recruitment, and plans to move away before delivery. The cohort profile was previously described by Oken and colleagues (*Oken et al., 2015*).

Current mid-childhood asthma was defined as a 'yes' response to 'Has a health professional ever told you that your child has asthma?' and 'yes' to either 'In the past 12 months, has child taken or been prescribed Albuterol, Cromolyn, Nedocromil, Montelukast, inhaled corticosteroids, or Prednisone' or 'In the past 12 months, has your child ever had wheezing (or whistling in the chest)?'. We used as a comparison group those with no asthma diagnosis and no asthma medication use or wheezing in the past 12 months. We used the same definition for current asthma in early adolescence, except the time reference for asthma medication was 'in the past month'. Ever asthma was defined as a 'yes' response to 'Has a health professional ever told you that your child has asthma?' within either the mid-childhood or early-teen follow-up periods. We used as a comparison group those with no asthma diagnosis. Individuals with missing data were not used in regression models assessing current or ever asthma outcomes. Individuals with missing data are reported in demographics tables and displayed in figure as 'Missing Data'.

Methods for obtaining spirometric measurements and BDR have been described previously (*Tse et al., 2013*). In brief, spirometry was performed with the EasyOne Spirometer (NDD Medical Technologies, Andover, MA). Post-bronchodilator spirometric measures were obtained at least 15 min after administration of two puffs (90 µg per puff) of albuterol. Spirometric performance was required to meet American Thoracic Society criteria for acceptability and reproducibility, with each subject producing at least three acceptable spirograms, two of which must have been reproducible (*Oken et al., 2015*; *Miller, 2005*).

## Human subject University of Illinois Hospital cohort

The current study was approved by the University of Illinois at Chicago IRB (#2015–0353). Women seeking prenatal care at the University of Illinois at Chicago (UIC) Center for Women's Health were recruited as volunteers in their third trimester (29–33 weeks of gestation) between 2014 and 2017. The cohort profile has been previously described by Koenig and colleagues (*Koenig et al., 2020*). Inclusion criteria were as follows: singleton pregnancy; naturally conceived pregnancy; 17–45 years of age; pre-pregnancy BMI $\geq$ 18.5; <34 weeks of gestation; sufficient fluency in English to provide consent and complete the study; and ability to independently provide consent. Exclusion criteria were as follows: live birth or another pregnancy (including ectopic and molar pregnancies) in the previous 12 months; pre-eclampsia; gestational diabetes mellitus or previously diagnosed type 1 or type 2 diabetes; autoimmune disorder; current or previous premature rupture of membranes or chorioamnionitis; previous spontaneous premature birth; current bacterial or viral infection; current steroid or anti-inflammatory treatment; history of bariatric surgery; malabsorptive condition (e.g., celiac disease); current hyperemesis; hematologic disorder (e.g., sickle cell anemia or trait, hemochromatosis); current tobacco use; alcohol consumption or illicit drug use; and current use of medications that decrease nutrient absorption (e.g., proton pump inhibitors). All women provided written informed consent.

## Umbilical cord blood specimens

For Project Viva, procedures for obtaining umbilical cord blood serum have been describe previously (*Schaub et al., 2005*). For the UIH cohort, umbilical cord blood was obtained by venipuncture shortly after time of delivery. Blood (approximately 5 mL per tube) was drawn into Red Top Serum Plus and Green Top Sodium Heparin 95 USP Units Blood Collection Tubes (BD Vacutainer). Red Tops were allowed to stand upright at room temperature for 30 min prior centrifugation at 1500 × g for 10 min at room temperature. Supernatants (serum) were collected, aliquoted, and stored at −80°C until further processing. Heparinized blood obtained in Green Tops was diluted 1:1 in 1× phosphate-buffered saline, pH 7.4 (PBS) and overlaid onto Ficoll-Paque Plus (GE Healthcare) density gradients. Density gradients were centrifuged at 400 × g for 30 min at room temperature without brake. Upper phase (diluted plasma) was drawn off, aliquoted, and stored at −80°C. Buffy coats (CBMCs) were drawn off, washed twice with 10 mL of 1× PBS, aliquoted, and stored in 500 mL of RNAlater (Qiagen). Viability and number of cells isolated were determined by diluting cellular suspensions 1:1 with Trypan Blue Solution 0.4% (wt/vol) in PBS (Corning) and counting live/dead cells > 7 µM using a TC20 Automated Cell Counter (Bio-Rad). The viability of isolated cells was >90% for all samples. Time from delivery to storage was recorded for every sample.

## RNA extraction and sequencing

Total RNA was extracted from CBMCs using RNeasy kits (QIAGEN) following manufacturers protocol except for switching 70% ethanol for 100% ethanol. The quality and quantity of all the extracted RNA were analyzed with a RNA 6000 Nano Kit on the 2100 Bioanalyzer Instrument (Agilent) and ssRNA High Sensitivity Kit and Qubit (Invitrogen). RIN for all samples was >8. RNA was constructed into barcoded libraries using the TruSeq Stranded mRNA Library Prep Kit (Illumina). The pooled libraries were sequenced for a paired-end 151 read length. The DNA libraries were sequenced on HiSeq X Ten platform using HiSeq Reagent v2.5 kit (Illumina), following manufacturer's protocol.

## Enzyme-linked immunosorbent assays (ELISA)

PGLYRP-1 and sIL6R$\alpha$ were assessed using Human PGLYRP1/PGRP-S DuoSet Elisa and Human IL-6 R alpha DuoSet Elisa (R and D Systems). Serum was diluted with 1% bovine serum albumin in PBS (pH 7.2–7.4, 0.2 micron filtered) at 1:100 for PGLYRP-1 and 1:300 for sIL6R$\alpha$. ELISAs were performed according to manufacturer's protocol. All samples were run in duplicate. Optical densities were assessed at 450 and 540 using a Spectra Max M5 (Molecular Devices). The intra- and inter-plater CVs for PGLYRP-1 were 2.4% and 11.0%. The intra- and inter-plater CVs for sIL6R$\alpha$ were 3.8% and 18.9%.

## Statistical analysis

All statistical analyses were performed in *R* (https://www.r-project.org/) unless otherwise specified.

## Meta-analysis

To identify dataset studies used in the meta-analysis, NCBI's Gene Expression Omnibus (https://www.ncbi.nlm.nih.gov/geo) was searched using the search '(cord blood) AND '*Homo sapiens*' [porgn:_txid9606]' and was limited to study types that included expression profiling by array. This search yielded 352 studies that were further examined for cell types assessed and meta-data reported. Seventeen studies met inclusion criteria of reporting metadata regarding at least one perinatal risk factor (e.g. gestational age at birth, newborn sex, birthweight, maternal pre-pregnancy BMI, smoke exposure, mode of delivery), and expression data from either whole cord blood or CBMCs derived from human subjects. Expression, feature, and subject demographic data were extracted using *geoquery* (*Davis and Meltzer, 2007*). If expression data was non-normalized, it was quantile normalized and log2 transformed. Six studies were excluded due to no variability in demographic data (i.e. only males) or low data quality, leaving 605 unique cord blood gene expression samples. To assess associations between gene expression and perinatal risk factors, univariate, inverse variance weighted, random-effects models were constructed for genes using the *GeneMeta* package (*GeneMeta, 2020*). Newborn sex (Male vs. Female), maternal pre-pregnancy BMI (continuous: 0 = BMI < 18.5, 1 = 18.5 $\leq$ BMI < 25, 2 = 25 $\leq$ BMI < 30, 3 = 30 $\leq$ BMI), gestational age at birth in weeks (continuous), and birthweight in grams (continuous) were assessed as perinatal risk factors. Significant genes were defined as Benjamini–Hochberg correct p-value<0.01. The z-score for each gene was averaged across each univariate test (*Equation 1*) and termed pooled z-score. It was used to assess how likely a gene is effected by multiple risk factors. If a gene was not assessed in one of the univariate analyses while assessed in others, the missing data was inferred as a z-score of zero. To determine cell enrichment, genes expression in each cell as defined by HPA (*Uhlen et al., 2010*; *Monaco et al., 2019*) were modeled as a function of the pooled z-score using general additive model with cubic splines function. To determine biological processes enriched in the low- vs. high-risk individuals, the pooled z-score was used a pre-ranked list for gene set enrichment analysis for GO biologic processes using *GSEA* (*Subramanian et al., 2005*). Significantly enriched GO terms were defined as Benjamini-Hochberg correct p-value<0.01. Spearman's correlations between pooled z-score, univariate z-scores, and individual dataset z-scores were determined in *R*.

$$Z_{pooled} = \frac{((Z_{Male} + Z_{PPBMI}) - (Z_{GA} + Z_{BW}))}{4} \quad (1)$$

## RNA sequencing statistical analysis (UIH cohort)

The sequences were quality controlled by filtering out all low-quality reads (<25 on Phred quality score) and short reads (<50 bp). Transcripts were annotated using *salmon v0.12.0* and *Ensembl*

*Homo sapiens* Genome Assembly GRCh38.12 (*Patro et al., 2017*). Transcript counts were aggregated in gene-level counts using the *tximport* package in R (*Soneson et al., 2015*). Genes with median counts across samples < 10 were filtered out, leaving 14,055 genes remaining whose expression normalized using median sum scaling. Normalized gene expression was modeled as function of the number of perinatal risk factors (gestational age < 37 weeks, birthweight < 3000 g, PP BMI >30, male) using *DESeq2* (*Love et al., 2014*). Genes were ranked for replication by their product of their pooled z-score and RNAseq z-score, termed RS (*Equation 2*). A cutoff of RS > 3 was used to determine candidate genes associated with pediatric asthma risk. Candidate gene biologic process and cellular component enrichment were performed using STRING with default settings (*Szklarczyk et al., 2019*). Data from RNAseq is available on NCBI SRA database ID PRJNA577955.

$$RS = Z_{pooled} \times Z_{RNAseq} \tag{2}$$

## Secondary statistical analysis of mass cytometry and ProSeek data (Olin et al.)

Methods for data acquisition for cell population percentages and protein abundances using mass cytometry and ProSeek are reported by *Olin et al., 2018*. Cell population percentages were transformed using centered log-ratios. Cell populations and protein abundances, and number of risk factors for each individual (gestational age < 37 weeks, male, birthweight < 3000 g, birth via c-section) were correlated (Pearson's method). For cell population correlations with number of risk factors, significance was defined as Bonferroni corrected p-value<0.05. Proteins determined to neutrophil associated were determined by those expressed in CBMC (UIH cohort) and enriched in neutrophils HPA (*Uhlen et al., 2010*).

## Asthma and pulmonary outcome statistical analysis (Project Viva)

Outcomes assessed in Project Viva Categorical outcomes were modeled using logistic regression. Continuous outcomes were modeled using linear regression. Four regression models were used for each outcome: univariate/unadjusted model 1 adjusted for child's demographics (gestational age at birth in weeks, birthweight adjusted for gestational age and sex, mode of delivery, sex, and race/ethnicity), model 2 adjusted for maternal demographics (PP BMI, race/ethnicity, level of education, atopy, antibiotic use during pregnancy, and smoking during pregnancy, 6 months, or 1 year), and model 3 adjusted for both mother and child's demographics (excluding maternal race/ethnicity). For regression models, PGLYRP-1 and sIL6Rα were log10 transformed and standardized to internal z-score. Subset analysis was performed by splitting the full data set by categorical variables and modeling outcomes as function of PGLYRP-1 in each subset using the univariate model. R (*Jaakkola et al., 2006*) for each variable in linear regression model 3 was determined using the *relaimpo* package (*Groemping, 2006*). Mcfadden's pseudo-R (*Jaakkola et al., 2006*) for each variable in model 3 was calculated for logistic regression using full models minus the variable of interest.

## Acknowledgements

Thanks to Kelly Liesse MD, Yishin Chang MS, and Wangfei Wang MS for their helpful comments regarding this manuscript.

## Additional information

### Funding

| Funder | Grant reference number | Author |
| --- | --- | --- |
| National Heart, Lung, and Blood Institute | F30HL136001 | Benjamin A Turturice |
| National Institute of Allergy and Infectious Diseases | R01AI053878 | Patricia W Finn |
| Eunice Kennedy Shriver National Institute of Child Health and Human Development | R01HD034568 | Emily Oken |

| National Institutes of Health | UH3OD023286 | Emily Oken |
| Robert Wood Johnson Foundation | Nurse Faculty Scholars Program #72117 | Mary Dawn Koenig |
| University of Illinois at Chicago | College of Nursing Dean's Award | Mary Dawn Koenig |

The funders had no role in study design, data collection and interpretation, or the decision to submit the work for publication.

## Author contributions

Benjamin A Turturice, Conceptualization, Resources, Data curation, Formal analysis, Funding acquisition, Validation, Investigation, Visualization, Methodology, Writing - original draft, Writing - review and editing; Juliana Theorell, Data curation, Validation, Investigation, Writing - review and editing; Mary Dawn Koenig, Resources, Data curation, Supervision, Funding acquisition, Validation, Investigation, Methodology, Project administration, Writing - review and editing; Lisa Tussing-Humphreys, Resources, Data curation, Supervision, Validation, Investigation, Methodology, Project administration, Writing - review and editing; Diane R Gold, Resources, Data curation, Funding acquisition, Investigation, Methodology, Project administration, Writing - review and editing; Augusto A Litonjua, Emily Oken, Resources, Data curation, Supervision, Funding acquisition, Investigation, Methodology, Project administration, Writing - review and editing; Sheryl L Rifas-Shiman, Data curation, Formal analysis, Methodology, Project administration, Writing - review and editing; David L Perkins, Patricia W Finn, Conceptualization, Resources, Data curation, Formal analysis, Supervision, Funding acquisition, Validation, Investigation, Methodology, Writing - original draft, Project administration, Writing - review and editing

## Author ORCIDs

Benjamin A Turturice (iD) https://orcid.org/0000-0001-9382-4612

## Ethics

Clinical trial registration NCT02820402.
Human subjects: The current study was approved by the University of Illinois at Chicago IRB, 20160326 and 20150353, and the IRB of Harvard Pilgrim Health Care.

## Decision letter and Author response

Decision letter https://doi.org/10.7554/eLife.63745.sa1
Author response https://doi.org/10.7554/eLife.63745.sa2

# Additional files

## Supplementary files

- Supplementary file 1. Meta-analysis of CBMC gene expression associated with newborn sex.
- Supplementary file 2. Meta-analysis of CBMC gene expression associated with gestational age.
- Supplementary file 3. Meta-analysis of CBMC gene expression associated with birthweight.
- Supplementary file 4. Meta-analysis of CBMC gene expression associated with maternal pre-pregnancy BMI.
- Supplementary file 5. UIH cohort demographics.
- Supplementary file 6. UIH cohort CBMC gene expression DESeq2 results.
- Supplementary file 7. Project viva demographics.
- Transparent reporting form

## Data availability

Data from RNAseq is available on NCBI Bioproject database PRJNA577955.

The following dataset was generated:

| Author(s) | Year | Dataset title | Dataset URL | Database and Identifier |
|---|---|---|---|---|
| Turturice BA, Theorell J, Koenig MD, Tussing-Humphreys L, Perkins DL, Finn PW | 2019 | Umbilical Cord Blood Mononuclear Cell mRNAseq: UIH Cohort | https://www.ncbi.nlm.nih.gov/bioproject/?term=PRJNA577955 | NCBI BioProject, PRJNA577955 |

The following previously published datasets were used:

| Author(s) | Year | Dataset title | Dataset URL | Database and Identifier |
|---|---|---|---|---|
| Mason E, Tronc G, Nones K, Matigian N, Kim J, Wolfinger R, Wells C, Gibson G | 2011 | Maternal influences on the transmission of leukocyte gene expression profiles in population samples (mother and child) | https://www.ncbi.nlm.nih.gov/geo/query/acc.cgi?acc=GSE21342 | NCBI Gene Expression Omnibus, GSE21342 |
| Dickinson P, Smith CL, Craigon M, Ross AJ, Khondoker MR, Forster T, Ivens A, Lynn DJ, Orme J, Jackson A, Lacaze P, Stenson BJ, Ghazal P | 2014 | Whole blood mRNA expression profiling of host molecular networks in neonatal sepsis | https://www.ncbi.nlm.nih.gov/geo/query/acc.cgi?acc=GSE25504 | NCBI Gene Expression Omnibus, GSE25504 |
| Votavova H, Merkerova Dostalova M, Fejglova K, Vasikova A, Krejcik Z, Tabashidze N, Veleminsky M Jr, Pastorkova A, Topinka J, Sram RJ, Brdicka R | 2011 | Comprehensive Study of Tobacco Smoke-Related Transcriptome Alterations in Maternal and Fetal Cells | https://www.ncbi.nlm.nih.gov/geo/query/acc.cgi?acc=GSE27272 | NCBI Gene Expression Omnibus, GSE27272 |
| Votavova H, Merkerova-Dostalova M, Krejcik Z, Fejglova K, Vasikova A, Tabashidze N, Pastorkova A, Topinka J, Balascak I, Sram RJ, Brdicka R | 2011 | Deregulation of Gene Expression induced by Environmental Tobacco Smoke Exposure in Pregnancy | https://www.ncbi.nlm.nih.gov/geo/query/acc.cgi?acc=GSE30032 | NCBI Gene Expression Omnibus, GSE30032 |
| Turan N, Sapienza C | 2012 | Genome-wide analysis of gene expression levels in placenta and cord blood samples from newborns babies | https://www.ncbi.nlm.nih.gov/geo/query/acc.cgi?acc=GSE36828 | NCBI Gene Expression Omnibus, GSE36828 |
| Joanna Dawn Holbrook | 2012 | Transcriptome changes affecting hedgehog and cytokine signalling in the umbilical cord in late pregnancy: implications for disease risk | https://www.ncbi.nlm.nih.gov/geo/query/acc.cgi?acc=GSE37100 | NCBI Gene Expression Omnibus, GSE37100 |
| Rager JE, Fry RC | 2014 | Prenatal arsenic exposure and the epigenome: altered gene expression profiles in newborn cord blood | https://www.ncbi.nlm.nih.gov/geo/query/acc.cgi?acc=GSE48354 | NCBI Gene Expression Omnibus, GSE48354 |
| Laajala E | 2014 | Standard of hygiene and immune adaptation in newborn infants | https://www.ncbi.nlm.nih.gov/geo/query/acc.cgi?acc=GSE53473 | NCBI Gene Expression Omnibus, GSE53473 |
| Edlow AG, Hui L, Wick HC, Bianchi DW | 2015 | The obese fetal transcriptome | https://www.ncbi.nlm.nih.gov/geo/query/acc.cgi?acc=GSE60403 | NCBI Gene Expression Omnibus, GSE60403 |
| Baldwin DA | 2016 | Unique inflammatory transcriptome profiles at the maternal fetal interface and onset of | https://www.ncbi.nlm.nih.gov/geo/query/acc.cgi?acc=GSE73685 | NCBI Gene Expression Omnibus, GSE73685 |

| | | | |
|---|---|---|---|
| | | human preterm and term birth | |
| Winckelmans E, Vrijens K, Tsamou M, Janssen BG, Saenen ND, Roels HA, Kleinjans J, Lefebvre W, Vanpoucke C, de Kok TM, Nawrot TS | 2017 | Newborn Sex-specific Transcriptome Signatures and Gestational Exposure to Fine Particles: Findings from the ENVIRONAGE Birth Cohort | https://www.ncbi.nlm.nih.gov/geo/query/acc.cgi?acc=GSE83393 | NCBI Gene Expression Omnibus, GSE83393 |

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
