## [Decision Letter]

**Acceptance summary:**

This study documents that the abundance of neutrophil specific granules at birth predicts the risk for pediatric asthma and reduced pulmonary function. The authors have used existing transcriptomic data from cord blood mononuclear cells, and have shown an inverse correlation between myeloid differentiation and the risk of pediatric asthma. This gene set was validated in an independent cohort and was found to contain associated genes localizing to neutrophil specific granules. Specifically, PGLYRP-1, a granule protein, was inversely associated with mid-childhood current asthma.

**Decision letter after peer review:**

Thank you for submitting your article "Perinatal Granulopoiesis and Risk of Pediatric Asthma" for consideration by *eLife*. Your article has been reviewed by two peer reviewers, and the evaluation has been overseen by Mone Zaidi as the Senior and Reviewing Editor. The following individual involved in review of your submission has agreed to reveal their identity: Michael Keane (Reviewer #3).

The reviewers have discussed the reviews with one another and the Reviewing Editor has drafted this decision to help you prepare a revised submission.

Summary:

This is a nice study that is clearly written and makes use of several datasets. It attempts to explain perinatal risk factors and the associated risk of developing pediatric asthma in the mid-childhood and early teenage years. Identified among maternal characteristics that were associated with risks of subsequent asthma development included atopy, BMI, race/ethnicity and demographics, birth characteristics, and mode of delivery. The paper then goes on to demonstrate the differences in immune response during the different time frames of pregnancy. Most notably, a gene signature associated with increased myelopoiesis in utero is associated with increased risk of pediatric asthma. Furthermore they show that cord blood serum PGLYRP -1 is associated with reduced risk of pediatric asthma and increased FEV1/FVC. Interestingly sIL6ra which is derived from neutrophils but not associated with neutrophil granules did not show any association with pulmonary outcomes. This suggests that it is the neutrophil granules rather than the neutrophils per se that are the problem association. The following should be addressed

Essential revisions:

1) Please readjust the Introduction; a lot of emphasis is placed on IFN/infection/asthma, but after this fact, it seems neglected. Furthermore, while the manuscript is clearly written, the message regarding PGLYRP-1 is at times confusing. There is clarity that PGLYRP -1 is inversely associated with mid childhood asthma risk. The Discussion however refers to animal models where PGLYRP -1 is proinflammatory and is associated with increased airway resistance and allergen sensitization. The apparent disparity should be clarified.

2) The take-home message for the paper – that there appears to be an inverse relationship between serum levels of PGLYRP-1 and overall risk for pediatric asthma – should be explored in relation to whether a predictive role for such proteins is possible since the authors can accurately predict risk factors for disease and assess pulmonary function. Other proteins, like the sIL6Ra, have no association with disease predictability and have no association with predicting pulmonary outcomes. This should be explored/explained in greater detail.

3) As part of the validation efforts of the study – the rationale for using three different cohorts to assess pediatric asthma risk was not clearly explained.

4) In terms of correlations between each protein level and pulmonary function – the sIL6Ra protein was not associated with the FEV1/FVC ratio or a bronchodilator response at either age group. However, it was found that increased levels of PGLYRP-1 were associated with an increased FEV1/FVC ratio (not indicative of asthma) and reduced odds of developing pediatric asthma at each age group. This analysis makes sense as increased production of neutrophil granules, PGLYRP-1, serves a protective effect against infection, reducing incidence of disease states. The paper, however, should explore the rationale behind the no-response to the sIL6Ra protein. In terms of understanding, since this protein is NOT associated with neutrophilic granules, it can be inferred, that is it may not have a role in protecting against infection. However, this could have been explored in more detail in the paper.

5) What is the proposed role of neutrophil degranulation in the pathogenesis or long term susceptibility to asthma?

---

## [Author Response]

Essential revisions:1) Please readjust the Introduction; a lot of emphasis is placed on IFN/infection/asthma, but after this fact, it seems neglected. Furthermore, while the manuscript is clearly written, the message regarding PGLYRP-1 is at times confusing. There is clarity that PGLYRP -1 is inversely associated with mid childhood asthma risk. The Discussion however refers to animal models where PGLYRP -1 is proinflammatory and is associated with increased airway resistance and allergen sensitization. The apparent disparity should be clarified.

We have edited the Introduction in response to the reviewer comments. We focus on known variability in the neonatal immune system that is associated with either asthma or perinatal risk factors for asthma. We have removed paragraphs regarding relationships between IFN and infection.

We have further edited and expanded our Discussion with regards to confusion concerning the role of PGLYRP-1 in asthma pathology. We indicated the relevance the murine *pglyrp-1*^-/-^ studies even though the methodologies of the studies limit the contextualization between those studies and ours. The murine studies have demonstrated a role for PGLYRP-1 in enhancing T2 airway inflammation in adult mice at the time of insult. Yet, these studies due to their methodologies have not been able to examine the impact of early life variation in PGLYRP-1 and its impact on asthma development. This highlights the temporal role of PGLYRP-1 in asthma which we have outlined in the Discussion.

2) The take-home message for the paper – that there appears to be an inverse relationship between serum levels of PGLYRP-1 and overall risk for pediatric asthma – should be explored in relation to whether a predictive role for such proteins is possible since the authors can accurately predict risk factors for disease and assess pulmonary function. Other proteins, like the sIL6Ra, have no association with disease predictability and have no association with predicting pulmonary outcomes. This should be explored/explained in greater detail.

In our revised Discussion, we have expanded upon the analyses of PGLYRP-1 and sIL6Rα. To reiterate, PGLYRP-1 is consistently associated with perinatal risk factors for asthma across cohorts and analyses, whereas sIL6Rα is not. Further, the stronger association between sIL6Rα, Black/African-American maternal race, and neutrophil abundance in CyTOF highlights that sIL6Rα is likely more reflective of absolute neutrophil count in cord blood. PGLYRP-1 is more associated with one aspect of functional diversity, secondary granule abundance, in neutrophils at birth which we had identified as a potential risk predictor for asthma through our meta-analysis. sIL6Rα was measured as a control and not expected to be associated with pulmonary outcomes. We have also detailed this rationale in our revised Results and Discussion.

3) As part of the validation efforts of the study – the rationale for using three different cohorts to assess pediatric asthma risk was not clearly explained.

We refer the reviewers to the Results section which state our rationale for the inclusion of each cohort. Specifically, we have detailed how each dataset cohort allows for examination of associations between either demographics or outcomes and various biologic components (mRNA, cells, serum proteins). Further, we underscore that we are able to reproducibly show the association of PGLYRP-1 with perinatal risk factors in each cohort and within our meta-analysis which advances the generalizability of our findings.

4) In terms of correlations between each protein level and pulmonary function – the sIL6Ra protein was not associated with the FEV1/FVC ratio or a bronchodilator response at either age group. However, it was found that increased levels of PGLYRP-1 were associated with an increased FEV1/FVC ratio (not indicative of asthma) and reduced odds of developing pediatric asthma at each age group. This analysis makes sense as increased production of neutrophil granules, PGLYRP-1, serves a protective effect against infection, reducing incidence of disease states. The paper, however, should explore the rationale behind the no-response to the sIL6Ra protein. In terms of understanding, since this protein is NOT associated with neutrophilic granules, it can be inferred, that is it may not have a role in protecting against infection. However, this could have been explored in more detail in the paper.

This reviewer comment is similar to the point raised in major concern 2. Please see our comment in response to major concern 2, as it address both concerns.

5) What is the proposed role of neutrophil degranulation in the pathogenesis or long term susceptibility to asthma?

Our objective was to identify biological pathways that are modulated by multiple risk factors to predict asthma in this observational study. We have stated that the lack of mechanism(s) due is a limitation in our Discussion. While investigated this mechanistic role(s) is of interest, it is beyond the scope of this study. We have discussed potential functional roles that PGLYRP-1 might play in early life immunity, but we refrain from proposing broader roles in asthma pathogenesis as this would be speculation.